# Engineering crystalline quasi-two-dimensional polyaniline thin film with enhanced electrical and chemiresistive sensing performances

Tao Zhang [1,2,10], Haoyuan Qi [3,10], Zhongquan Liao[4,10], Yehu David Horev[5], Luis Antonio Panes-Ruiz [6], Petko St. Petkov [7,8], Zhe Zhang[2,9], Rishi Shivhare[2,9], Panpan Zhang[1,2], Kejun Liu [1,2], Viktor Bezugly[6], Shaohua Liu[1,2], Zhikun Zheng [1,2], Stefan Mannsfeld [2,9], Thomas Heine [2,7], Gianaurelio Cuniberti [2,6], Hossam Haick[5], Ehrenfried Zschech[2,4], Ute Kaiser[3], Renhao Dong[1,2] & Xinliang Feng[1,2]

Engineering conducting polymer thin films with morphological homogeneity and long-range molecular ordering is intriguing to achieve high-performance organic electronics. Polyaniline (PANI) has attracted considerable interest due to its appealing electrical conductivity and diverse chemistry. However, the synthesis of large-area PANI thin film and the control of its crystallinity and thickness remain challenging because of the complex intermolecular interactions of aniline oligomers. Here we report a facile route combining air-water interface and surfactant monolayer as templates to synthesize crystalline quasi-two-dimensional (q2D) PANI with lateral size ~50 cm$^2$ and tunable thickness (2.6–30 nm). The achieved q2D PANI exhibits anisotropic charge transport and a lateral conductivity up to 160 S cm$^{-1}$ doped by hydrogen chloride (HCl). Moreover, the q2D PANI displays superior chemiresistive sensing toward ammonia (30 ppb), and volatile organic compounds (10 ppm). Our work highlights the q2D PANI as promising electroactive materials for thin-film organic electronics.

[1] Faculty of Chemistry and Food Chemistry, Technische Universität Dresden, 01062 Dresden, Germany. [2] Center for Advancing Electronics Dresden (cfaed), Technische Universität Dresden, 01062 Dresden, Germany. [3] Central Facility of Electron Microscopy, Electron Microscopy Group of Materials Science, Universität Ulm, 89081 Ulm, Germany. [4] Fraunhofer Institute for Ceramic Technologies and Systems (IKTS), Maria-Reiche-Strasse 2, 01109 Dresden, Germany. [5] Department of Chemical Engineering and Russell Berrie Nanotechnology Institute, Technion-Israel Institute of Technology, Haifa 32000003, Israel. [6] Institute for Materials Science and Max Bergmann Center of Biomaterials and Center for Advancing Electronics Dresden (cfaed), Technische Universität Dresden, 01062 Dresden, Germany. [7] Wilhelm-Ostwald-Institute of Physical and Theoretical Chemistry, Leipzig University, Leipzig, Germany. [8] University of Sofia, Faculty of Chemistry and Pharmacy, Sofia, Bulgaria. [9] Department of Electrical and Computer Engineering, Technische Universität Dresden, 01062 Dresden, Germany. [10] These authors contributed equally: Tao Zhang, Haoyuan Qi, Zhongquan Liao. Correspondence and requests for materials should be addressed to X.F. (email: xinliang.feng@tu-dresden.de)

The discovery of linear conducting polymers made of organic monomers[1] has led to excitement over their potential applications such as chemical (bio)sensors[2], optical displays[3], solar cells[4], organic light-emitting diodes[5], transistors[6], and supercapacitors[7]. It is well known that structural disorder hinders efficient charge transport in conducting polymer films[8,9], thus degrades device performance. To achieve long-range charge transport, one promising strategy is to align the linear conducting polymer chains into quasi-two-dimensional (q2D) crystalline films[10,11]. The q2D film, composed by highly ordered supramolecular assembly of molecules/polymers with fully expanded-coil conformation via interchain interactions[12–14], can provide multiple pathways for interchain charge transport[8,9] and bypass possible defects of individual polymer chains[15].

In the grand family of conductive polymers, polyaniline (PANI) has been most studied owing to its outstanding electrical, magnetic, and optical properties[16–19]. Vigorous effort has been devoted to fabricating PANI thin films, including (i) top-down solution processing of PANI chains via spin coating[20], drop casting[21], and Langmuir-Blodgett technology[22], and (ii) bottom-up synthesis from monomers via self-assembled monolayer (SAM) templating[23], chemical vapor deposition[24,25], and air (or liquid)-liquid interfacial method[26,27]. However, these strategies only produced inhomogeneous and amorphous (i.e., randomly compact-coil conformation) PANI films or partially crystalline nanofiber-, rod- and sphere-shaped PANIs[21,22,26], due to the poor processability of PANIs and complex intermolecular interactions of aniline/oligomers[28]. Therefore, the conductivity of resultant PANI thin films (using HCl as dopant) is typically below 1 S cm$^{-1}$ [21–23,26]. Moreover, molecular-level structure of the reported PANI films has not yet been resolved.

Despite recent advance in the synthetic methodologies, e.g., templating by solid crystals[29,30], graphene[31] and surfactant bilayers[32], as well as pyrolysis of single crystal solids[33], the synthesis of crystalline q2D PANI films with long-range order remains a significant challenge. Here, we report a synthesis of crystalline q2D PANI films with wafer-scale size (~50 cm$^2$) and tunable thickness from 2.6 nm to 30 nm by combining an air-water interface (i.e. confined reaction environment) and surfactant monolayer (i.e., soft crystalline template) templating strategy. The resultant q2D PANI films have crystalline grains with sizes up to ~2.3 µm. Within each grain, the PANI chains align in a perfect expanded-coil conformation along the lateral direction of the film. The q2D PANI manifests anisotropic charge transport characteristics with an intrinsic lateral conductivity of $8.7 \times 10^{-3}$ S cm$^{-1}$ and a vertical conductivity of $5.0 \times 10^{-5}$ S cm$^{-1}$. Subsequent vapor-phase HCl doping remarkably enhances the lateral conductivity to ~160 S cm$^{-1}$, which is the highest value for PANI thin films (e.g., thickness <30 nm) reported to date. The ultra-thinness in conjunction with high crystallinity render q2D PANIs high-performance electrode materials for chemiresistive sensors, enabling sensitive detection of ammonia gas down to 30 ppb and volatile organic compounds (e.g., heptanal) at 10 ppm.

## Results

### Synthesis and morphology

The q2D PANI film was synthesized via the oxidative polymerization of aniline monomers at the air-water interface with the assistance of a surfactant monolayer. The synthesis procedure is schematically illustrated in Fig. 1a. Surfactant monolayer (e.g., sodium oleyl sulfate) was firstly prepared on water surface in a glass well (50 mL) with a diameter Ø = 6 cm, followed by the addition of aniline monomers (11.5 µL in 1 mL water) in the water subphase. The glass well was then covered with a petri dish and kept for ~24 h, allowing aniline

monomers to diffuse and adsorb underneath the surfactant monolayer (Supplementary Fig. 1)[34]. Afterwards, 1 M HCl (1 mL) and ammonium persulfate (APS, 10 mg in 1 mL water) were added sequentially into the subphase triggering the oxidative polymerization of aniline at 1 °C (Fig. 1b, c). The polymerization was slowed down by using low concentration of monomer and oxidant (i.e., APS), which could be favorable for the formation of ultra-thin PANI films with high crystallinity. After ~48 h polymerization, a uniform and continuous q2D PANI film was obtained on the water surface.

Using the above synthetic strategy, both air-water interface and surfactant monolayer are key factors for determining the formation of crystalline q2D PANI films: (i) they facilitate simultaneous self-assembly and polymerization of aniline monomers into ordered polymer chains under the anionic head groups of surfactant monolayer via hydrogen bonding and electrostatic interactions; (ii) they provide a confined environment (between surfactant monolayer and water surface) for the thin film formation; (iii) free oligomers and polymers (in solution) that cannot interact with the surfactant monolayer would precipitate, and thus do not participate in the film formation.

To transfer the q2D PANI film, a solid substrate was placed under the floating film and the water subphase was removed gently until the film fell onto the substrate surface (Supplementary Fig. 2). Q2D PANI film with a diameter of ~8 cm could be fully transferred onto a 300 nm thick SiO$_2$/Si wafer (diameter Ø = 10 cm, Fig. 2a). Under optical microscope, the q2D PANI is uniform, and the edges of the film are clearly visible (Fig. 2b). The q2D PANI can suspend over large holes with edges of ~20 µm on a copper grid (Fig. 2c), suggesting a high mechanical stability. Atomic force microscopy (AFM) measurement at film edges by stochastic sampling reveals an average thickness of ~9.3 nm after 48 h of polymerization (Fig. 2d). The thickness is nearly identical at different positions, and the root mean square (RMS) roughness of selected area ($5 \times 5$ µm$^2$) is 0.3 nm, indicating excellent morphological homogeneity of the q2D PANI film.

To demonstrate the crucial role of the surfactant monolayer, various surfactants with different head groups and hydrophobic chains (Supplementary Fig. 3) were investigated. The morphologies of the q2D PANIs derived from various surfactant monolayers were inspected by optical microscopy (Supplementary Fig. 4). The utilization of cationic and nonionic surfactants (e.g., octadecylamine, hydrogen ionophore IV and lignoceryl alcohol) leads to rough PANI films, while anionic surfactants (e.g. sodium oleyl sulfate and sodium dodecylbenzenesulfonate) produce large-area continuous and uniform PANI films. Moreover, applying sulfate ions headed surfactants affords q2D PANI films with excellent morphological homogeneity without cracks and pinholes (Supplementary Fig. 4). This can be attributed to the highest negative charge density of the sulfate groups, which facilitates the electrostatic interaction with aniline monomers (Fig. 1c and Supplementary Fig. 3)[23,32,34]. Without using surfactant monolayer, only fibrous PANI was formed (Supplementary Fig. 5).

### Crystal structure and domain size

We employed selected area electron diffraction (SAED) and aberration-corrected high-resolution transmission electron microscopy (AC-HRTEM) to probe the crystallinity and molecular structure of q2D PANI. Highly reproducible SAED patterns have been observed from the free-standing q2D PANI thin film (~9.3 nm thick; Fig. 3a and Supplementary Fig. 6), demonstrating its excellent crystallinity. Based on the rectangular symmetry and absences of odd-order h00 and 0k0 reflections (i.e., p2gg plane group), the unit cell parameters are determined as: $a = 6.8$Å, $b = 7.4$Å, and $\gamma = 90°$.

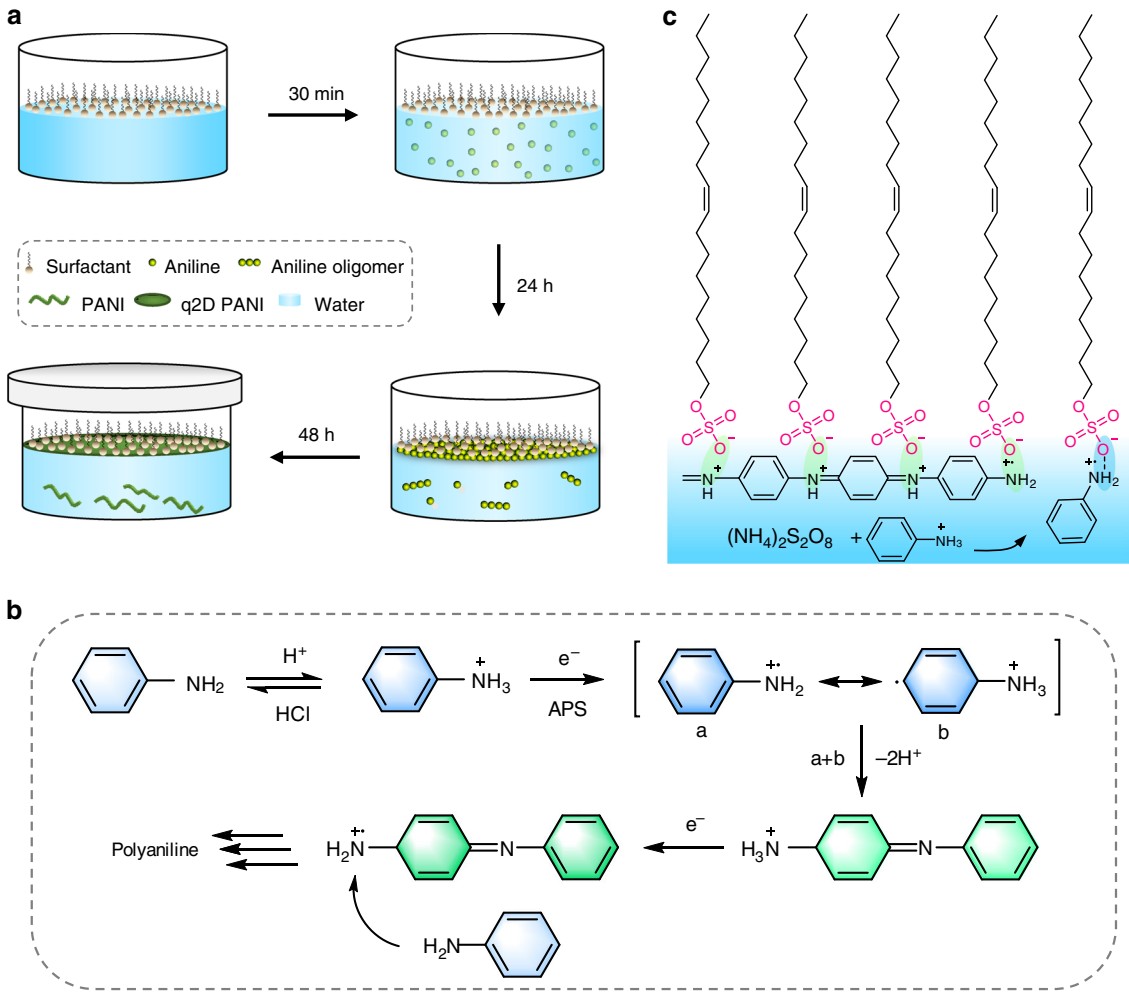

**Fig. 1** Synthetic of q2D PANI and reaction mechanism. **a** Schematic illustration of the synthetic procedure of q2D PANI: preparation of surfactant monolayer on the water surface; addition of aniline into water subphase and standing for 24 h for the diffusion of monomers to the water subphase and interface; introduction of HCl and APS to the water subphase; oxidative polymerization for 48 h. **b** The mechanism of oxidative polymerization of aniline. **c** Schematic demonstration of the hydrogen bonding (blue ellipse) and electrostatic interaction (green ellipse) between protonated aniline/oligomer cations and sulfonate group of sodium oleyl sulfate

The statistical value of single crystal domain size derived by SAED is 1.1–1.5 μm (i.e., 1.2–2.3 μm², Supplementary Fig. 7). Remarkably, the largest crystalline domain size is beyond 2.3 μm (i.e., ~5.2 μm²), which is substantially larger than that of crystalline PANI obtained on ice surface (~68 nm)[29]. Interestingly, similar to the highly ordered alkanethiolate SAMs obtained on Au[111][35], the misorientation between adjacent PANI domains is typically below 15° (Supplementary Fig. 8), implying low defect density in the q2D PANI thin film[35,36]. When no or cationic/neutral surfactants were applied (e.g., octadecylamine, hydrogen ionophore IV, lignoceryl alcohol) (Supplementary Fig. 9), only amorphous or partially crystalline PANI films were obtained.

The molecular structure of q2D PANI was visualized by AC-HRTEM imaging. As shown in Fig. 3b and Supplementary Fig. 10, the linear polymer chains align parallel to each other, packing into a q2D molecular sheet. Unlike polymers obtained by solution synthesis[37], the PANI chains in the molecular sheet exhibit excellent long-range order, showing no chain folding or any entanglement. Since the average single crystal size of q2D PANI is 1.1–1.5 μm, we estimate that the length of the PANI chains in each crystal reaches the same scale, corresponding to ~10⁶ monomer units and ~10⁸ g mol⁻¹ molecular weight in a single PANI chain. Such molecular weight is about three orders of

magnitudes higher than that prepared from solution synthesis (~10⁵ g mol⁻¹)[38].

Well-defined layer structure of q2D PANI crystal is revealed by SAED and AC-HRTEM acquired perpendicularly to the [001] axis, which demonstrates an interplanar spacing of 13.5Å (Fig. 3c and Fig. 3d, and Supplementary Fig. 11). Furthermore, grazing-incidence wide-angle X-ray scattering (GIWAXS) performed on a q2D PANI film (~30 nm thick) on SiO₂/Si wafer discloses a monoclinic unit cell with, $a = 6.79$ Å, $b = 7.45$ Å, $c = 13.41$ Å, and $\alpha = 97°$, $\beta = \gamma = 90°$ (Supplementary Figs. 12 and 13). The absence of odd-order h00 and 0k0 reflections further verifies the p2gg plane group symmetry. From the AC-HRTEM and GIWAXS results, the molecular structure of the q2D PANI can be resolved and depicted as shown in Fig. 3e. The adjacent chains along b direction are opposite to each other with an edge-on π-π stacking of polymer chains. Calculated 2D model of q2D PANI and corresponding SAED patterns are in agreement with the experimental results (Fig. 3f, Supplementary Fig. 14).

**Thickness control and spectroscopic characterization**. The q2D PANI formation is confined at the surfactant-water interface, in which the monomers in the water subphase continuously

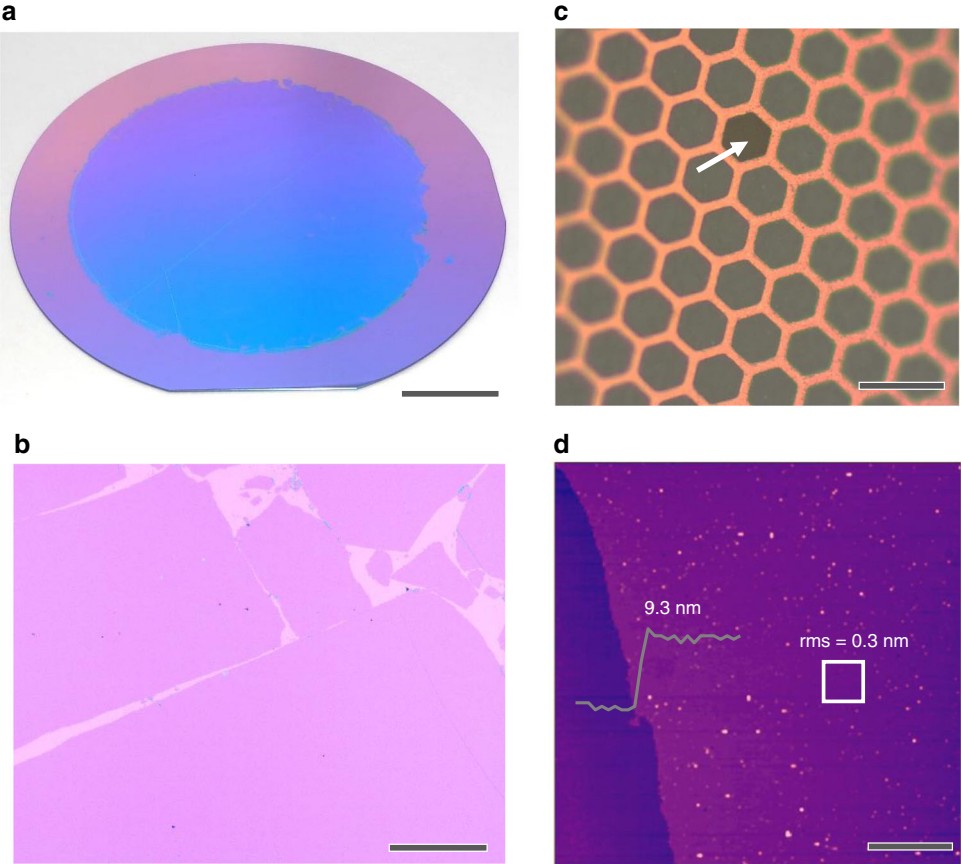

**Fig. 2** Morphology of the q2D PANI film. **a** q2D PANI on a 300 nm SiO₂/Si wafer (diameter Ø = 10 cm). The reaction time is 48 h. The uniform color indicates that the film (diameter Ø = 8 cm) is homogeneous. **b** Optical microscopy image of q2D PANI. **c** Freestanding q2D PANI on a copper TEM grid. The white arrow points to a hole in the q2D PANI film, which is in contrast to surrounding freestanding film. **d** Atomic force microscopy (AFM) image and height profile of q2D PANI. The RMS roughness was measured in a selected area of $5 \times 5\,\mu m^2$ marked by the white box. Scale bars: **a** 2 cm; **b** 40 μm; **c** 200 μm; **d** 10 μm

transport to the interface for oxidative polymerization. Therefore, extending the reaction time leads to a higher monomer conversion, corresponding to an increase in film thickness. As revealed in Fig. 4a, the thickness of q2D PANI increased with a constant rate of $\delta d = 5$ nm per day (in 0.02 M HCl) in the first five days, then levelled around 30 nm after seven days when all monomers were consumed (Supplementary Fig. 15). In order to increase the doping level of q2D PANI, the acid concentration increased to 1 M during polymerization, while a longer induction period[39] (~12 h) was observed and the polymerization speed decreased to 4.2 nm per day (Supplementary Fig. 16). Notably, the film crystallinity improved substantially with increasing thickness, and the crystal structure remained identical (Supplementary Figs. 17 and 18). The thickness of the thinnest q2D PANI film was $2.6 \pm 0.4$ nm (corresponding to two molecular layers, one layer is ~1.3 nm according to the above SAED and GIWAXS results), which was obtained after a 12 h reaction (Supplementary Figs. 19–21).

From ultraviolet–visible–near-infrared (UV–Vis–NIR) spectra (Supplementary Fig. 22), the q2D PANI presents the characteristic absorbance at 430 nm (polaron-π⋆)[40], which shows a linear correlation with reaction time in the initial five days (Fig. 4b), and follows the Beer-Lambert law[41]. The transmittance of q2D PANI decreases with reaction time (Fig. 4b). Nevertheless, ~90% of transmittance can still be observed on the q2D PANI after 7 days of reaction (~30 nm thick) which can be attributed to the excellent chain ordering that reduces light scattering (Fig. 3a)[42]. By increasing the HCl concentration of water subphase from 0.02 to 1 M, we can identify a monotonic rise in the absorbance at 360

nm (π–π⋆ transition of the benzenoid ring) and above 600 nm (free-carrier absorption) (Fig. 4c), which are characteristics of the doped form of PANI (emeraldine-salt)[43] and beneficial for achieving high electrical conductivity.

**Conductivity measurement.** The electrical conductivity of the as-prepared q2D PANI films was measured by two-probe (lateral conductivity) and current-sensing AFM (vertical conductivity), respectively (Supplementary Fig. 23). The corresponding I-V curves along both directions indicate an ohmic contact between −0.5 V and +0.5V, which reveal a lateral conductivity of $8.7 \times 10^{-3}$ S cm⁻¹ (red line in Supplementary Fig. 23c) and a vertical conductivity of $5.0 \times 10^{-5}$ S cm⁻¹ (black line in Supplementary Fig. 23c) in a 9.3-nm-thick q2D PANI film doped by 0.02 M HCl. The superior lateral conductivity ascribes to the long-range ordered and expanded-coil conformation of PANI chains along the in-plane direction, which enhances hopping transport between adjacent PANI chains[13,14,44]. In contrast, the PANI counterparts prepared at air-water interface without and with cationic or non-ionic surfactant monolayers (e.g., octadecylamine, hydrogen ionophore IV, lignoceryl alcohol) present much lower conductivity values ($<8.3 \times 10^{-7}$ S cm⁻¹, Supplementary Table 1).

When the doped acid concentrations of subphase increased from 0.02 to 1 M, the corresponding lateral conductivity of q2D PANI increased to 23 S cm⁻¹ (Fig. 4d and Supplementary Fig. 24). The I-V current (0.69 mA at 50 mV) of q2D PANI doped with 1 M HCl is even superior to the commercial graphene (0.61 mA at

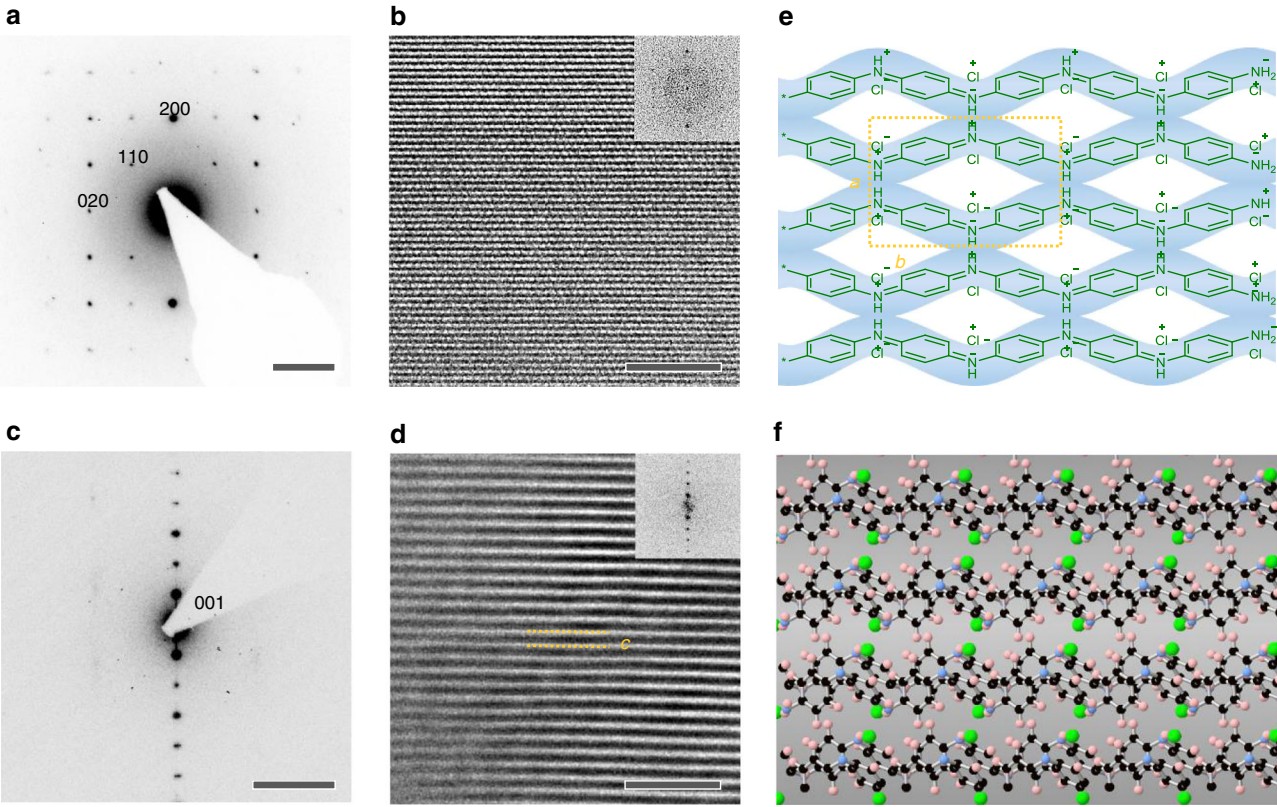

**Fig. 3** Structural characterization of q2D PANI single crystal. **a** SAED pattern and **b** AC-HRTEM image of q2D PANI along [001] axis. The 200 and 020 reflections are at 2.96 nm$^{-1}$ and 2.70 nm$^{-1}$, respectively. Inset of **b** corresponding FFT. **c** SAED and **d** AC-HRTEM image of q2D PANI perpendicular to [001] axis. The two yellow lines mark out the interlayer distance $c = 13.41$Å. Inset of **d** corresponding FFT. **e** Schematic illustration of the stacking of linear PANI chains into q2D PANI. The yellow rectangle marks out the unit cell in [001] direction, where $a = 6.79$Å and $b = 7.45$ Å. **f** Simulated atomic structure of the q2D PANI. Scale bars: **a** 2 nm$^{-1}$; **b** 5 nm; **c** 2 nm$^{-1}$; **d** 10 nm

50 mV) synthesized by chemical vapor deposition. The conductivity of q2D PANI increased to 160 S cm$^{-1}$ by additional doping using HCl vapor (Supplementary Fig. 25). It is worth noting that such conductivity of the q2D PANI largely surpasses those of reported PANI thin films of low crystallinity (Supplementary Table 2; Supplementary Figs. 26 and 27).

**Chemical sensing**. Owing to their ultra-thinness and wide range tunability of electrical conductivity (e.g., upon exposure to acid, alkali and polar compounds), the q2D PANI is a promising electrode material for chemical sensing[2,45]. The performance of q2D PANI in NH$_3$ sensing was firstly assessed through a chemiresistor-type gas sensor, which was fabricated through transferring a 9.3-nm-thick q2D PANI onto SiO$_2$ substrate covered with Au electrodes (Supplementary Fig. 28). Figure 5a shows the normalized sensing response $\Delta R/R_0$ to successive exposures to NH$_3$ with concentrations ranging from 15 to 120 ppb under room temperature. In all tested devices, a decrease of current (an increase of resistance) upon NH$_3$ exposure was observed, which is due to the deprotonation of q2D PANI by NH$_3$[46]. The lowest detection limit (defined as the concentration providing a signal-to-noise ratio of at least 3)[47] was 30 ppb, lower than the most reported PANI sensors (Fig. 5b and Supplementary Table 3). Such sensitivity is even better than nitrogen- and boron-doped carbon nanotubes (100 ppb) at identical testing conditions[48], and relevant for diagnosis of certain diseases such as live cirrhosis, kidney failure, and diseases caused by Helicobacter pylori[49]. The high performance of q2D PANI in NH$_3$ sensing can be attributed to its ultra-

thinness with the sufficient exposure of activity sites as well as long-range ordered chain structures that provide efficient pathways for the diffusion of NH$_3$ molecules (~1.2 Å).

Next, the potential application of q2D PANI film in clinical related chemiresistor was evaluated by exposure to volatile organic compounds (VOCs) (Supplementary Fig. 29). Heptanal, as a representative VOCs, has been detected in blood, breath, and urine samples[50], and thus could serve as a biomarker for disease diagnosis and health monitoring[51,52]. Figure 5c displays the sensing characteristics of the 5-nm-thick q2D PANI based chemiresistor, which reveals extremely fast response after exposure to heptanal vapor and the excellent reversibility when flushed with dry nitrogen. The electrical resistance of the chemiresistor increases with the rise of heptanal concentrations (from 10 to 50 ppm), which is likely caused by the swelling of q2D PANI from the heptanal (polar) adsorption. The physical/weak binding between VOCs and q2D PANI ensures a reversible (peak-like) resistance response in sensing. In addition, sensitivity can be modulated by various doping acids as well as the film thickness (Fig. 5d). Overall, the ~2% $\Delta R/R_0$ (at 10 ppm) of q2D PANI rivals the state-of-art PANI based devices (~1.7 % at 25 ppm)[53], and sufficient to detect the VOCs released from patients (~205.5 ppm) and healthy controls (~22.8 ppm)[54]. In comparison, lower sensitivity of the q2D PANI chemiresistors corresponds to a lower polarity VOCs (3-heptanone) (Supplementary Fig. 30). The above sensing experiments suggest that the q2D PANIs have considerable potentials for the fabrication of sensors for gas sensing and clinical applications.

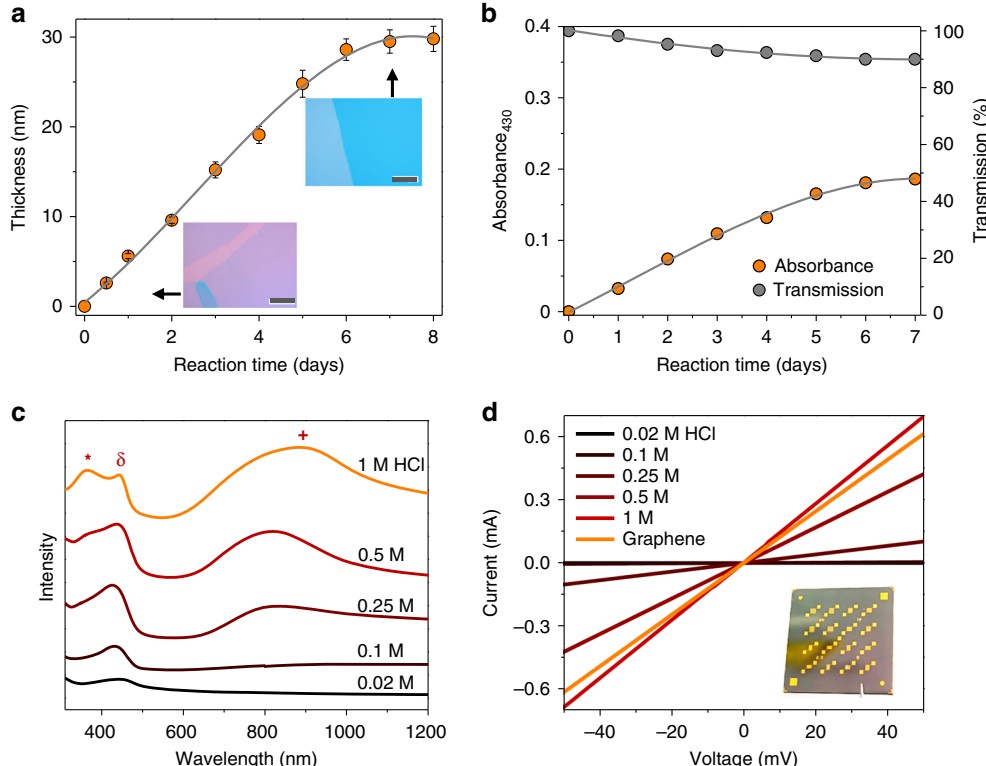

**Fig. 4** Spectroscopic and electrical conductivity characterizations. **a** Thickness of q2D PANI vs. reaction time. Inset: optical microscopy images of q2D PANI in 1 day and 7 days, respectively. Error bars indicate the variations in thickness of each q2D PANI sample at five different positions. Scale bars: 50 μm. **b** Plot of 430 nm absorbance and corresponding transmittance of q2D PANIs from (**a**). **c** UV–Vis–NIR absorption of q2D PANI prepared at various HCl acid concentrations from 0.02 to 1 M. **d** I-V characteristic curves of q2D PANI from (**c**), and in comparison to graphene-CVD. Inset: photograph of q2D PANI on commensal organic field-effect transistor substrate for I-V measurement

## Discussion

We report a methodology for the synthesis of crystalline q2D PANI thin films with large lateral size ~50 cm$^2$ and tunable thickness (2.6–30 nm). The achieved q2D PANI exhibits single crystal domain size as large as 2.3 μm and lateral electrical conductivity (for PANI thin films) up to 160 S cm$^{-1}$. Furthermore, the q2D PANI displays ability to detect ammonia gas as low as 30 ppb as well as volatile organic compounds (VOCs) at 10 ppm (e.g., heptanal).

In consideration of the broad interest of PANI, we expect that the q2D PANI thin films will find many other applications in such as transparent electrodes, flexible supercapacitors and functional membranes. Given that oxidative polymerization is applicable to the synthesis of other conducting polymers, we anticipate that crystalline q2D thin films of polypyrrole, polythiophene and their analogues can also be developed to enhance electrical properties and device performances.

## Methods

**Materials**. Octadecylamine, hydrogen ionophore IV, lignoceryl alcohol, stearic acid, sodium stearate, dihexadecyl phosphate, sodium dodecylbenzenesulfonate, sodium oleyl sulfate, ammonium persulfate, chloroform, dimethyl sulfoxide, hydrochloric acid (37%) and PANI emeraldine base (50,000 g mol$^{-1}$) were purchased from Sigma-Aldsh. Perfluorooctadecanoic acid and aniline (>99%) was obtained from Alfa Aesar. All the chemicals were used as received. Single layer CVD graphene was purchased from Graphene Supermarket Inc.

**Synthesis of q2D PANI**. Each surfactant was dissolved in chloroform (1 mg mL$^{-1}$) and filtered by PFTE syringe filter (0.2 μm, 1–15 mL, ThermoFisher). 10 μL of the surfactant solution was spread on a water (50 mL, Millipore) surface in glass well (60 mL), and allowed chloroform to evaporate for 30 minutes. 1 mL of aniline solution (11.5 μL, 0.13 mM in water) was added gently to the subphase using a pipette. After standing for 24 h for the diffusion of monomers in the water

subphase and interface, 1 mL of HCl (0.02–1 M in water) and 1 mL ammonium persulfate (APS, 10 mg, 0.044 mM, in water) were added to the subphase in 30 min, respectively. The glass well was then covered by a glass slide and placed in a refrigerator (Liebherr FKUv 1660 Premium, Germany) at 1 °C for the oxidative polymerization. The solution turned eventually dark green after several hours, indicating the successful polymerization of aniline monomers into PANI. q2D PANI subsequently appeared on the water and were fished using arbitrary substrates. Before characterization, the q2D PANI films on substrates were rinsed for ~1 h with chloroform (20 mL) and ethanol (20 mL), respectively.

**Synthesis and transfer of wafer-scale q2D PANI**. The wafer-scale of q2D PANI was synthesized in a 150 mL crystallising dish (diameter ∅ = 8 cm), and 100 mL Millipore water was used. The synthetic procedures are same to above mentioned, while the amounts of applied reagents (e.g., aniline, APS and HCl) were doubled.

The transfer procedures for the wafer-scale q2D PANI are shown in Supplementary Fig. 2. In brief, the glass well was placed in a larger one with diameter ∅ = 15 cm, followed by adding water till that the q2D PANI film floated onto the water surface. Afterwards, a 300 nm SiO₂/Si wafer (4-inch, Microchemicals GmbH) was placed under the film, and the water was removed slowly using a plastic dropper. As such, the q2D PANI dropped slowly to the wafer surface. The resultant q2D PANI on SiO₂/Si was washed with chloroform and ethanol, and then dried in vacuum oven.

**Isotherm of surfactant monolayer**. A Langmuir-Blodgett trough (Minitrough, KSV NIMA, Finland) equipped with a platinum Wilhelmy plate, a taflon dipper and a pair of delrin barriers, was used to measure the surface pressure-mean molecular area (π-A) isotherm of surfactant monolayer. Chloroform solution (150 μL) of the surfactant (sodium oleyl sulfate, 1 mg mL$^{-1}$) was spread onto pure water subphase with a microsyringe. After 30 min, when the solvent was evaporated, the π-A isotherm was recorded at a continuous pressing speed for the barrier of 1 mm min$^{-1}$ at room temperature. In addition, aniline (0.23 μL mL$^{-1}$) and aniline hydrochloride desolated (0.23 μL mL$^{-1}$) water were also used as subphase to spread the surfactant, and π-A isotherms were measured respectively for comparison.

**Controlled PANI films prepared by spin-coating**. The PANI films of different thickness on 300 nm SiO₂/Si were prepared by spin coating (4000 r.p.m. with 30 S) from different concentrations of PANI emeraldine base (5 × 10$^4$ g mol$^{-1}$) in

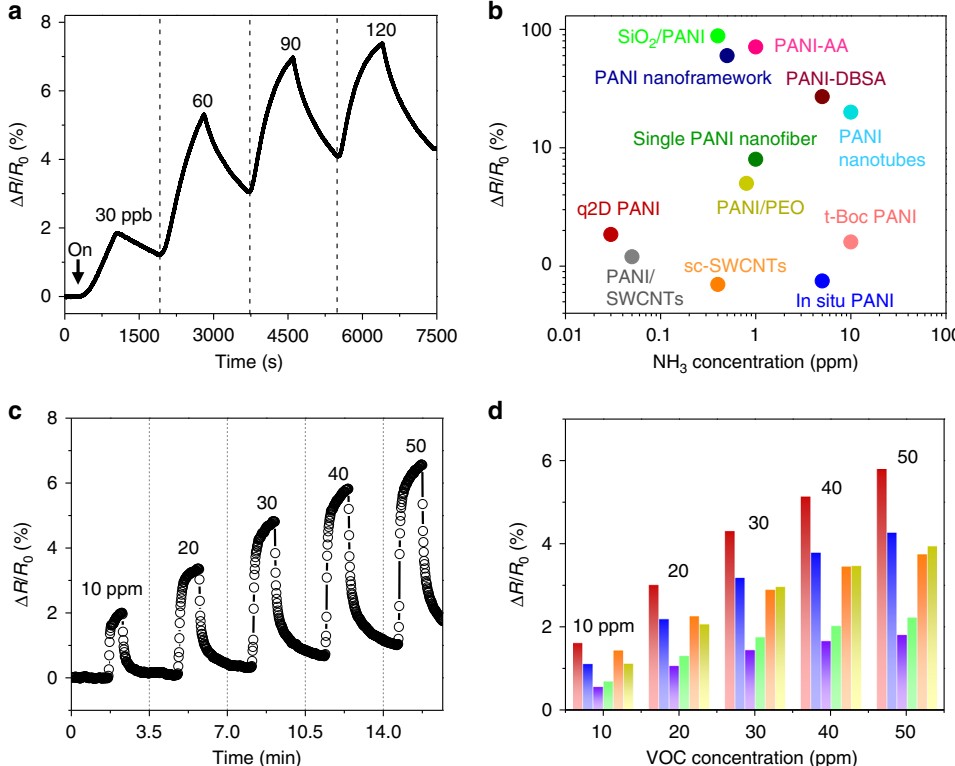

**Fig. 5** Ammonium and volatile organic compounds (VOCs) chemiresistor. **a** The sensing response ($\Delta R/R_0$) of q2D PANI (1 M HCl) to various ammonia concentrations. **b** Plot of $\Delta R/R_0$ vs. ammonia concentration of q2D PANI in comparison to other reported PANI-based sensors. **c** Sensing response $\Delta R/R_0$ of q2D PANI (0.02 M) chemiresistor under exposure to different heptanal concentrations of 10, 20, 30, 40, and 50 ppm. **d** Column diagram of sensor arrays to heptanal based on q2D PANI with various dopants: 0.02 M HCl (~5 nm, red); 0.02 M HCl (~9.3 nm, blue); 0.005 M HCl (~9.3 nm, violet); 0.02 M sulfuric acid (~9.3 nm, green); 0.02 M phytic acid (~9.3 nm, orange); 0.02 M trifluoromethanesulfonic acid (~9.3 nm, yellow)

DMSO. The PANI film of 6.1 nm thick was prepared by spin coating from a 5 mg mL$^{-1}$ PANI solution, and 11.3 nm from 10 mg mL$^{-1}$ and 20.8 nm from 10 mg mL$^{-1}$ via twice spin-coating processes. The thickness was measured with variable angle spectroscopic ellipsometry at room temperature. All films were doped in HCl vapor before the conductivity measurement.

**Electrical measurements**. Resistance ($R$) of q2D PANI film was measured by the Jandel cylindrical probe combined with the RM3000 test unit, and the electrical conductivity ($\sigma$) was determined by the equation, $\sigma = 1/\rho = 1/R*d$, where R ($\Omega$ sq$^{-1}$) is sheet resistance. The thickness ($d$) of q2D PANI was determined by AFM (NT-MDT) with tapping mode. The current-sensing AFM (CS-AFM) technique was used to measure the vertical conductivity of q2D PANI on Au/Si wafer. The characterization was performed with an Agilent AFM 5420 (USA) in contact mode using standard Au coated tips with a curvature radius of 10 nm and under ambient conditions. The *I–V* curves of the q2D PANI (on commercial organic field-effect transistor substrate, Fraunhofer IPMS) were measured with 2.5 μm source-drain channel at ambient conditions using a commercial Lakeshore Hall System.

**NH$_3$ sensor fabrication**. Cr/Au interdigitated electrodes (IDE) were fabricated on p-type Si substrates by standard photolithography, thermal evaporation and lift-off processes. The substrates were cleaned with acetone, isopropanol and DI water in a bath sonicator. Then, the q2D PANI on water surface was directly transferred on the IDE area using above mentioned method. The electrodes pad and measurement system were connected, using silver paste and co-axial conducting wire.

**NH$_3$ sensing experiment**. Q2D PANI based sensors were exposed to various concentrations of NH$_3$ for 15 min separated by 15 min of recovery under pure N$_2$ flow at room temperature in a self-designed gas exposure system (Supplementary Fig. 28). Different NH$_3$ concentrations were achieved by diluting NH$_3$ gas in pure N$_2$ using two mass flow controllers (MFC). A constant voltage ($V_{SD}$) of 0.1 V was applied between electrodes and the change in source-drain current ($I_{SD}$) was read using a Keithley 2602 source meter. The normalized sensing response is defined as the relative resistance change and was calculated using Eq. (1).

$$\text{Response}(\%) = \frac{\Delta R_t}{R_0} = \frac{I_0 - I_t}{I_t} \times 100 \qquad (1)$$

where $\Delta R_t$ is the difference in resistance before and during NH$_3$ exposure, $R_0$ and $I_0$

are the values of resistance and current before NH$_3$ exposure, and $I_t$ the current at a fixed voltage which is monitored during gas exposure experiments.

**VOC sensor fabrication**. A 300 nm SiO$_2$/Si wafer was cleaned with acetone, ethanol, and DI water in sequence before being blown dry in N$_2$. Then, Ti (5 nm) and Au (100 nm) were evaporated onto the SiO$_2$/Si by e-beam evaporation under the protection of a silicon mask in sequence. Then, the substrates were cleaned with acetone, isopropanol and DI water in a bath sonicator. The q2D PANI on water surface was directly transferred on the electrodes area using above-mentioned method. The electrodes pad and measuring system were connected, using silver paste and co-axial conducting wire.

**VOCs sensing experiment**. For VOCs sensing, vapors of all analytes were generated by a computer-controlled bubbler system and the temperature of analytes are well controlled by a water bath, for above 1ppm concentration; for low concentrations, gas-generator was used, along with permeation oven, to control the analytes temperature. Nitrogen was used as a carrier gas as well as a reference gas. The total flow rate was kept in 10 L min$^{-1}$ during the experiment. All vapor concentration of analytes are calculated as reported[55]. Exposures (5 min) were carried out after 10 min nitrogen baseline or recovery. All of the response measurements were carried out using a Keithley 2636A system SourceMater and a Keithley 3706 system Switch/Multimeter (Supplementary Fig. 29). All measurements were taken place in stainless steel chamber.

**Crystal domain size analysis**. The single-crystalline domain size of q2D PANI was evaluated by the consecutive acquisition of SAED patterns coupled with well-defined specimen stage movement. Initially, the selected-area aperture was centered in the field of view (Supplementary Fig. 6a) for the acquisition of the first SAED pattern at this position (Supplementary Fig. 6b). Subsequently, the specimen stage was shifted laterally by 0.4 μm where the second SAED pattern was recorded (Fig. S6c). The shift-and-acquire procedure was consecutively carried out until a series of position-dependent SAED patterns have been obtained (Supplementary Fig. S6d–f). This technique is equivalent to moving the selected area along the horizontal direction in 0.4 μm steps within the specimen plane, allowing us to estimate the single-crystalline domain size (Supplementary Fig. 7).

**Other characterizations**. The morphology and structure of the 2D-PANI nanosheet samples were investigated by transmission electron microscopy (Zeiss, Libra 200 kV), scanning electron microscopy (SEM, Zeiss Gemini 500), and optical microscopy (Zeiss) with a Hitachi KP-D50 color digital CCD camera. High-resolution TEM images was taken on an image-side aberration-corrected FEI Titan 80-300 transmission electron microscope operated at an acceleration voltage of 300 kV. XPS measurements were performed on a PHI-5000C ESCA system with a monochromatic Mg Kα X-ray source (hv = 1253.6 eV), the C 1 s value was set at 284.6 eV for charge corrections. Raman spectra and maps were measured on a NT-MDT confocal spectrometer with a 532 nm laser, and the spot size of the laser beam was ~0.5 μm. Infrared spectra were recorded on a FT-IR Spectrometer Tensor II (Bruker) with an ATR unit. Atomic force microscopy (AFM) images were recorded in air on a customized Ntegra Aura/Spectra from NT-MDT (Moscow, Russia) with a SMENA head in semicontact mode. The probes have a typical curvature radius of 6 nm, a resonant frequency of 47-50 kHz, and a force constant of 0.35–6.10 N m$^{-1}$. Height determination and calculation was performed with the software Nova Px 3.2.5 from NT-MDT and the free software Gwyddion. The structure of q2D PANI was calculated using tight binding density functional approach with DFTB+package[56] and 3ob-3-1 parameters data set[57,58]. Corresponding SAED patterns of the modeled structure was simulated using CrystalMaker.

## Data availability

The data that support the findings of this study are available from the corresponding author on reasonable request.

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

## Acknowledgements

This work was financially supported by the ERC Grant T2DCP, ESF Young Researcher Group 'GRAPHD' and the EC under the Graphene Flagship (no. CNECTICT-604391). The German Excellence Initiative via the Cluster of Excellence EXC1056 "Center for Advancing Electronics Dresden" (cfaed) is gratefully acknowledged. We acknowledge Petr Formánek (Leibniz-Institut für Polymerforschung Dresden) for TEM. We would like to thank M. Schwartzkopf for assistance (in using P03-MINAXS beamline, DESY). The authors thank Xiaodong Zhuang for valuable discussions.

## Author contributions

X.F. and T.Z. conceived and designed the experiments. T.Z. performed the synthesis. H.Q., U.K., Z.L., and E.Z. performed AC-HRTEM imaging, SAED and corresponding analysis. Y.D.H. and H.H. performed the VOCs sensor test and analysis. P.S.P. and T.H. performed DFT calculation. L.A.P.-R., V.B., and G.C. performed the NH$_3$ sensor test and analysis. Z. Zhang performed the conductivity measurement. R.S., K.L., and S.M. performed the GIWAXS measurement and data analysis. P.Z., S.L., R.D., and Z. Zheng rendered characterization and helpful discussion. All authors discussed the results and commented on the manuscript.

## Additional information

**Competing interests:** The authors declare no competing interests.

