## [Peer Review File · Nature Communications]

Reviewers' comments:

Reviewer #1 (Remarks to the Author):

In this paper, large-area PANI thin film with controlled crystallinity and thickness was synthesized with the help of anionic surfactant (i.e. olyelsulfate) at water/air interface. The novelty of this paper is the long-range molecular ordering and high chemiresistive sensing performance of PANI thin films. The idea of growth 2D nanomaterial at water/air interface with the confine of ionic surfactants has already been developed in ref. 31(Nat. Commun. 2016, 7, 10444). Here, the authors extended the method to synthesize conducting polymer and obtained PANI thin film with controlled thickness at the range of 2.6 nm to 30 nm, and large single crystal domain size. The crystal structures of the film with thickness of 30 nm was well analyzed. Interestingly, this film shows very high lateral conductivity after doping with HCl vapor and excellent performance towards the sensing of ammonia and volatile organic compound (like heptanal), which will highlight the applications of PANI for thin-film organic electronics. This work is of interest to researchers in the relevant field. A few issues, as following, should be addressed:

1. The surfactant monolayer provides a confined environment for the thin film formation. The anionic head groups of surfactant monolayer have the critical role for facilitating the polymerization of aniline monomers into ordered polymer chains via hydrogen bonding and electrostatic interactions. Does the packing density of the monolayer have any effect on the morphology of the PANI film? Can the crystallinity of the film be further improved by tuning the packing density of the monolayer? In page3, diameter of 6 cm glass well was used for the growth of PANI. After transferred onto SiO₂/Si wafer, the authors claimed PANI film with diameter of 8 cm can be obtained. The size of the film is larger than the reaction container. Was the film synthesized in the same container?
2. The thickness for film after 48 h polymerization is uniform, and the root mean square roughness is 0.3 nm which confirms the morphological homogeneity. How about the thinner and/or thicker film (i.e. 2.6-nm or 30-nm film)? do they still show the same morphological homogeneity?
3. Does the crystal structure and the crystal domain size of the film show thickness dependence? During the film thickness growth, is there any evolution of the crystallinity? Authors have showed SAED patterns for 9.3-nm PANI film and AC-HRTEM images for 30-nm PANI film. Is there any difference of the crystalline domain size between these two films? It would have been good, if authors could provide the crystallinity information for thinner PANI film, such as 2.6-nm or 5-nm film.
4. In Fig. 4a, the thickness of q2D PANI increased with constant rate of $\delta d = 6$ nm per day in the first five days. However, the thickness of the film at 5 days in the plot is around 25 nm.
5. Owing to the high crystallinity, the PANI film with thickness of 9.3 nm exhibits anisotropic charge transport when doped with 0.02 M HCl. Does the anisotropic charge transport property still maintain after doped with HCl vapor?
6. Response time is also an important factor for evaluating the performance of chemiresistor sensor. The author claims that the film shows extremely fast response after exposure of heptanal vapor. It would be better to quantify how fast the response is in the text. How about the response time of PANI for NH₃ sensing comparing with the other reported PANI-based sensors?

Reviewer #2 (Remarks to the Author):

Review is attached

Reviewer #3 (Remarks to the Author):

General Comments

This paper reports a simple method using air-water interface and surfactant monolayer as

templates to synthesize crystalline quasi-two-dimensional (q2D) PANI and It can be tunable thickness (2.6-30 nm). In addition, the authors demonstrate that crystal structure quasi-two-dimensional (q2D) PANI by grazing-incidence wide-angle X-ray scattering (GIWAXS) and Aberration-corrected high-resolution transmission electron microscopy (AC-HRTEM). Due to high crystallinity, q2D PANI exhibits ultra-high lateral conductivity up to 160 S cm⁻¹ doped by HCl and superior chemiresistive sensing toward ammonia as low as 30 ppb, and volatile organic compounds (e.g. heptanal) at 10 ppm. However, The authors needs to explains the relationship between tunable thickness and crystallinity. Besides, the manuscript is strewn with poor usage of the English language. Therefore, I recommend publication of the manuscript with a MAJOR REVISION, in light of the following specific comments:

Specific Comments

1. The authors stated that free oligomers and polymers (in solution) that cannot interact with the surfactant monolayer will precipitate. Figure 1 (a) also shows this content. Then it means that there is a limit to film formation? In other words, The authors should explain exactly principle of the thickness control of the film.
2. The authors stated that low concentrations of monomer are beneficial. Proper reasons should be presented. Do you also control the concentration of monomer for thickness control of the film? If so, why is there a precipitated polymers?
3. After addition of aniline into water sub phase, The authors add hydrochloric acid and ammonium persulfate after 24h. What is the clear reason to put in after 24 hours? Proper reasons should be presented.
4. After 48h, Atomic force microscopy (AFM) image show homogeneity film thickness. then Is film thickness still uniform and homogeneity after 7 days? In addition, The authors have to show whether optical microscopy images also appear uniform as a function of time, In other words, The authors should explain that film has uniform and continuous in various film thickness through Atomic force microscopy (AFM) image or optical microscopy image.
5. In Figure 1, The authors have to make a clear notation in the picture. Make it clearly what is the oligomers, polymer, APS, HCl and so on.
6. In Figure 3, AC-HRTEM image is appeared clearly, but check again Scale bars. (b) and (d) represent 5nm and 10nm respectively. Is it correct? Then The authors should show low magnification of TEM image formed 2D single crystal.
7. The manuscript requires a complete overhaul of the language (grammar, missing articles, inappropriate phrasing, etc.). The authors are strongly encouraged to take the support of native language editors for the purpose.

Reviewer #1 (Remarks to the Author):**Comments:**

In this paper, large-area PANI thin film with controlled crystallinity and thickness was synthesized with the help of anionic surfactant (i.e. olyelsulfate) at water/air interface. The novelty of this paper is the long-range molecular ordering and high chemiresistive sensing performance of PANI thin films. The idea of growth 2D nanomaterial at water/air interface with the confine of ionic surfactants has already been developed in ref. 31 (Nat. Commun. 2016, 7, 10444). Here, the authors extended the method to synthesize conducting polymer and obtained PANI thin film with controlled thickness at the range of 2.6 nm to 30 nm, and large single crystal domain size. The crystal structures of the film with thickness of 30 nm was well analyzed. Interestingly, this film shows very high lateral conductivity after doping with HCl vapor and excellent performance towards the sensing of ammonia and volatile organic compound (like heptanal), which will highlight the applications of PANI for thin-film organic electronics. This work is of interest to researchers in the relevant field. A few issues, as following, should be addressed:

Response:

We greatly appreciate for the valuable comments from the reviewer. All the suggestions and comments from the reviewer have been carefully addressed and modifications have been made accordingly.

Question 1:

The surfactant monolayer provides a confined environment for the thin film formation. The anionic head groups of surfactant monolayer have the critical role for facilitating the polymerization of aniline monomers into ordered polymer chains via hydrogen bonding and electrostatic interactions. Does the packing density of the monolayer have any effect on the morphology of the PANI film? Can the crystallinity of the film be further improved by tuning the packing density of the monolayer? In page3, diameter of 6 cm glass well was used for the growth of PANI. After transferred onto SiO₂/Si wafer,

the authors claimed PANI film with diameter of 8 cm can be obtained. The size of the film is larger than the reaction container. Was the film synthesized in the same container?

Response:

We thank the reviewer for the thoughtful comments. We fully agree that the packing density of the surfactant monolayer strongly influences the morphology of the PANI film. Actually, the use of surfactant monolayer as template for biomineralization and mineralization at air-water interface has been widely studied around 1990s (*Science* 261, 1286, 1993; *Nature* 328, 63, 1987; *Nature* 334, 692, 1988). It has been demonstrated that surfactant monolayer with lower packing density (or surface pressure) can promote better nucleation of molecules/salts than that of higher packing density (*J. Am. Chem. Soc.* 1998, 120, 2090-2098; *Langmuir* 1997, 13, 7165-7172). This is because the nucleation beneath the monolayer is mainly governed by two factors: 1) strong adsorption on the monolayer; 2) a correspondence between the structure/geometry of the monolayer and the nucleating crystal face.

Therefore, the monolayer of low packing density has a high tolerance to the lattice mismatches between the monolayer film and the nucleating face, which therefore results in high crystallization (*J. Am. Chem. Soc.* 1998, 120, 2090-2098; *Langmuir* 1997, 13, 7165-7172). In our work, various surfactant monolayers have been investigated in the synthesis of q2D PANI (Supplementary Figs. 3 and 4). Compared with the other surfactants, the sodium oleyl sulfate (SOS) always results in the q2D PANI with the highest crystallinity and uniformity. The superiority of SOS can be attributed to the lower surface pressure or low packing density of SOS monolayer at the air-water interface, as revealed by the surface pressure-mean molecular area (π - A) isotherm measurement (Supplementary Fig. 1). Although the SOS monolayer has been intensively compressed using barriers in a Langmuir-Blodgett trough, there was no solid phase formed at the air-water interface, which is in contrast to other surfactants such as stearic acid that prefers to form solid-state monolayer (Figure R1; *Nature* 334, 692, 1988).

Figure R1 | Surface pressure-mean molecular area (π -A) isotherms of stearic acid monolayer.

In another scenario, when the amount of SOS at the air-water interface decreases to 2 μL (1 mg/mL), it is too low to form a continuous monolayer (according to initial isotherm test, the minimum amount of SOS to form a continuous SOS monolayer on water surface with diameter $\varnothing = 6$ cm is ca. 7 μL). As a consequence, we could only observe rough/discontinuous PANI film (Figure R2b). When the amount of SOS increased to 7 μL or even more, the resultant PANI film became highly uniform and continuous (Figure R2c). We have included the new results and discussion into revised Supplementary Fig. 5.

Figure R2 | **Optical microscopic images of PANI films.** The films were prepared at (a) the pure air-water interface, (b) air-water interface with 2 μL (1 mg mL⁻¹) SOS, (c) air-water interface with 7 μL (1 mg mL⁻¹) SOS. Scale bars: 20 μm .

We are sorry for the confusion on the diameters of the glass wells. The wafer-sized q2D PANI film was synthesized in a glass well of 8 cm in diameter (Supplementary Fig. 2); all of other films were prepared in the glass well of 6 cm. We have modified the text in the revised manuscript. In addition,

we added an additional paragraph in Supplementary Method to describe the synthesis of wafer-sized q2D PANI film.

Question 2:

The thickness for film after 48 h polymerization is uniform, and the root mean square roughness is 0.3 nm which confirms the morphological homogeneity. How about the thinner and/or thicker film (i.e. 2.6-nm or 30-nm film)? do they still show the same morphological homogeneity?

Response:

The morphological homogeneity is slightly different for the films with different reaction time/thickness. As revealed by AFM scans (Supplementary Fig. 19), the thinner film of 2.6 nm shows a lower roughness of ca. 0.2 nm. In this revision, we measured thicker film of ca. 30.8 nm (seven days reaction; Figs. R3a and R3b), which showed a higher roughness of ca. 0.9 nm. Nevertheless, the thicker film is still quite homogeneous at microscopic scale as revealed by optical microscopy images (Fig. R3c), which is in clear contrast to that prepared in the absence of surfactant monolayer (Fig. R2a). We have included this updated results into revised Supplementary Fig. 15.

Figure R3 | q2D PANI film of 30.8 nm in thickness. (a) AFM topographic image of q2D PANI film. The area of black box was selected to calculate root-mean-square (RMS) roughness. (b) Corresponding height profile of the grey line in (a). (c) Optical microscopic image of q2D PANI film. Scale bars: (a) 10 μm ; (c) 20 μm . The film was prepared with seven days reaction at air-water interface with SOS surfactant monolayer.

Question 3:

Does the crystal structure and the crystal domain size of the film show thickness dependence? During the film thickness growth, is there any evolution of the crystallinity? Authors have showed SAED patterns for 9.3-nm PANI film and AC-HRTEM images for 30-nm PANI film. Is there any difference of the crystalline domain size between these two films? It would have been good, if authors could provide the crystallinity information for thinner PANI film, such as 2.6-nm or 5-nm film.

Response:

We thank the reviewer for the constructive comments. In order to address the dependence of the crystal structure and crystal domain size on the film thickness, we performed SAED measurements on the q2D PANI films collected at different reaction times (from 12 h to 7 days). As shown in Figure R4, the crystal structure of the films are always identical at various thicknesses. However, the crystallinity continuously and obviously increased during the reaction/polymerization, since the SAED patterns become clearer and brighter.

Previously we calculated the crystalline domain size of 9.3 nm thick film using SAED (Supplementary Fig. 7). In the revised manuscript, we further measured the domain size of thicker ca. 30.8 nm film using the same method. As revealed in Figure R5, the dominant size of crystal domains has increased to 1.5-1.9 μm in the q2D PANI film of 30.8 nm thickness (in comparison to 1.1-1.5 μm domain size with 9.3 nm thick film; see Supplementary Fig. 7).

Figure R4 | SAED patterns of PANI films with different thicknesses. (a) 2.6 nm (12 h); (b) 5.8 nm (ca. 24 h); (c) 10.5 nm (ca. 48 h); (d) 19.6 nm (4 days); (e) 30.8 nm (7 days).

The above new results have been included in the Supplementary Figs. 17.

Figure R5 | Histogram of domain size in the q2D PANI of 30.8 nm thick. The result was derived from 55 domains by SAED.

The results of above histogram of crystal domain size have been included in the Supplementary Fig. 18.

Question 4:

In Fig. 4a, the thickness of q2D PANI increased with constant rate of $\delta d = 6$ nm per day in the first five days. However, the thickness of the film at 5 days in the plot is around 25 nm.

Response:

We are sorry for this mistake, and thank the reviewer for pointing this out. In the revision, we have corrected the growth rate of δd to 5.0 nm per day.

Question 5:

Owing to the high crystallinity, the PANI film with thickness of 9.3 nm exhibits anisotropic charge transport when doped with 0.02 M HCl. Does the anisotropic charge transport property still maintain after doped with HCl vapor?

Response:

Following the reviewer's suggestion, we have performed the lateral and vertical conductivity measurements on the same sample after additional doping with HCl vapor from concentrated HCl (37%) in a sealed flask for 2 h. As shown in Figure R6, the anisotropic conductivity maintains after doping.

Figure R6 | Electrical conductivity characterization of the q2D PANI film after HCl doping. (a) Lateral $I-V$ characteristic curve of the q2D PANI film from two probe method. (b) Vertical $I-V$ characteristic curve of the q2D PANI from CS-AFM. A lateral conductivity $\sigma_L = 11.5 \text{ S m}^{-1}$ and vertical conductivity $\sigma_V = 4.6 \times 10^{-4} \text{ S m}^{-1}$ were obtained respectively. The PANI film is same to that of used in Supplementary Figs. 23.

Question 6:

Response time is also an important factor for evaluating the performance of chemiresistor sensor. The author claims that the film shows extremely fast response after exposure of heptanal vapor. It would be better to quantify how fast the response is in the text. How about the response time of PANI for NH_3 sensing comparing with the other reported PANI-based sensors?

Response:

We thank the reviewer for the constructive comment. The response time for the heptanal vapor at 10 ppm is 0.13 min (ca. 8 s; Fig. R7). Following the reviewer's suggestion, we have compared the response time of q2D PANI to NH_3 with other reported PANI-based sensors (Table R1). The response time of q2D PANI at 30 ppb is about 11.2 min. It is worth to note that the response time can be affected significantly by the sensing testing setup (e.g. the direction and distance of the NH_3 outlet to sample surface as well as the flow rate of the gas). Using the same NH_3 sensing testing setup, the response time of q2D PANI based chemiresistor sensor (ca. 11.2 min for 30 ppb) is superior to the state-of-art single-walled carbon nanotubes based sensors (ca. 15 min for 400 ppb; ACS Sens. 2018, 3, 79).

Figure R7 | Response time of q2D PANI based chemiresistor in sensing heptanal. The figure is derived from the Fig. 5c in the manuscript.

Table R1 | Literature reported high-performance PANI-based NH₃ sensors. The polyaniline-inorganic composites based sensors are excluded.

Polyaniline samples	[NH₃] ppm	Sensitivity (%)	Response time (min)	Refs.
PANI-HCl nanofibers	100	~2400	~25	Nano Lett. , 2004 , 4, 491.
PANI nanofibers	100	~2900	~25	J. Am. Chem. Soc. , 2003 , 125, 314.
PANI nanotubes	10	~20	~33	Macromol. Rapid. Commun. 2007 , 28, 286.
t -Boc PANI nanofibers	10	1.6	~16	Macromol. Mater. Eng. 2016 , 301, 1320.
PANI-DBSA thin film	5	~27	2	Sci. Adv. Mater. 2015 , 7, 518.
In situ PANI	5	~0.75	~1	J. Mater. Chem. C , 2015 , 3, 9461.
PANI-AA	1	71	2.5	Sens. Actuators B Chem. , 2001 , 77, 657.
Single PANI nanofiber	1	~8	~0.15	Sensors , 2011 , 11, 6509.
PANI-coated MWNTs	1	~5	4	Mater. Sci. Eng. B , 2009 , 163, 76.
PANI/PEO nanowire	0.8	~5	0.16	Nano Lett. 2004 , 4, 671.
PANI nanoframework	0.5	~60	~1	Nano Lett. 2004 , 4, 1693.
SiO ₂ /PANI nanofibers	0.4	88	~9	Colloids Surf. A , 2018 , 537, 532.
Thin-film-sc-SWCNT	0.4	~1	15	ACS Sens. , 2018 , 3, 79.
PANI/SWCNTs	0.05	~1.2	~20	Electroanalysis 18 , 2006 , 12, 1153.
PANI mesh	0.0025	~2.6	2.5	RSC Adv. , 2018 , 8, 5312.
q2D PANI	0.03	1.85	11.2	This work

The item of *Response time* of PANI-based NH₃ sensors has been included in Supplementary Table 3.

Reviewer #2 (Remarks to the Author):

Comment:

This work deals with the formation of highly crystalline 2D PANI films using surfactant approach resulting in conductivity up to 23 S/cm (by *in situ* HCl doping), which can be further enhanced up to 160 S/cm under external HCl doping. The as-formed PANI films are further shown to act as VOC sensors. The fact that the authors can synthesize large-area 2D conducting polymer films with high crystallinity is interesting. However, extremely long synthesis time weakens the benefit and novelty of current approach. This reviewer has several concerns and comments on this manuscript, as detailed below.

Response:

We greatly thank the reviewer for the valuable comments on our manuscript. All the suggestions and concerns from the reviewer have been carefully addressed and modifications have been made accordingly.

Question 1:

The role of air–water interface in the synthesis is unclear. The surfactants preferably stay at water surface (not in the bulk solution) as they are added below their critical micelle concentration. In addition, authors have stated “aniline monomers diffused and adsorbed underneath the amphiphilic surfactant monolayer (Supplementary Fig. 1).” The surfactant layer resists air contact to surfactant-bound monomers present in water phase. Also, the free monomers (unbound to surfactant) simply precipitate and does not participate in film formation (line 102).

Response:

The air-water interface facilitates the formation of a surfactant monolayer with hydrophobic alkyl chains facing to the air and hydrophilic head groups to the water face, which then serves as a soft template guiding the crystallization of PANI. As illustrated in Fig. 1c, the free monomers/oligomers that could not bound to surfactant monolayer will precipitate (Fig. 1a), which does not affect the formation of the q2D PANI film at the air-water interface.

Question 2:

Surfactant is not removed after polymerization. The residual surfactant should make impact on the evaluated properties and characterization of as-formed PANI films.

Response:

We are sorry for this confusion. Indeed we have removed the surfactant after polymerization. Before characterization, the surfactant has been removed by rinsing the specimens in chloroform (20 mL) and ethanol (20 mL) for ca. 1h. The sample after washing was checked by XPS, and the sulphur content (originating from the SO₄⁻ head group of SOS) is below 0.1% (Fig. R8), which is within the limit of the detection. Therefore, we can conclude that there is no surfactant impact on the evaluated properties of the PANI films. Corresponding description about the purification procedures has been included in the Supplementary Methods (Page S30): “Before characterization, the q2D PANI films on substrates are rinsed for ca. 1 h with chloroform (20 mL) and ethanol (20 mL), respectively.”

Figure R8 | X-ray photoelectron spectroscopy (XPS) survey spectrum of q2D PANI after washing.

Question 3:

Recent literature demonstrated a few min synthesis of 2D PANI film (*Angew. Chem.* **2015**, *54*, 10501) and other 2D conjugated polymer film (<https://pubs.acs.org/doi/10.1021/acsnano.8b07294>) using ice-air interface. Proposed method certainly revealed shortcomings in this aspect, but this issue is not addressed in the manuscript.

Response:

We thank the reviewer for sharing these two papers. We are aware of these two excellent works reported previously, which presented a highly interesting approach to synthesize q2D PANI and PEDOT:PSS films. The reason for the rather slow reaction/polymerization in our approach is due to the low concentrations of monomer (0.13 mM) and oxidant (APS, 0.044 mM), which is much lower

than the ice-templating method using 0.25 M monomer and oxidant (*Angew. Chem.* 2015, 54, 10501). The use of low concentrations of reactants in this work aimed to decrease the rates of nucleation, polymerization and crystal growth, and thus to achieve the synthesis of ultra-thin (e.g. < 30 nm) PANI films of excellent crystallinity. Therefore, our current work has clear different motif from the literature efforts.

Considering the reviewer's suggestion, we have addressed this issue in the revised manuscript (Page 3 Line 23) "*The polymerization was slowed down by using low concentration of monomer and oxidant (i.e. APS), which could be favorable for the formation of ultra-thin PANI films with high crystallinity.*" Moreover, the *Angew. Chem.* paper (*Angew. Chem.* 2015, 54, 10501) was cited in our previous version, and the *ACS Nano* paper has been cited in the revised manuscript as Ref. 28.

Question 4:

In order to polymerize aniline, 50 mL of reactant mixture is stored from 2 to 7 days with a varied concentration of HCl from 0.02 to 1 M in water. Isn't there occurring any freezing of water under long time polymerization at 0 °C (line 499, Supplementary file)? In addition, the increase in concentration of HCl also causes depression of freezing point of water. However, the temperature condition statement is ambiguous. For example, authors have mentioned polymerization temperature to be 0 °C in main manuscript (line92) while it is stated to be 1°C in the Supplementary file (line 499). This is critical while validating the migration of reactants to the surface to make a film. There are chances to generate typical grain boundaries within ice depending on reaction temperature and concentration of HCl. This issue is not addressed in the manuscript.

Response:

We appreciate the reviewer for the constructive comment. The reaction temperature for the whole polymerization was kept at 1°C rather than 0°C. We are sorry for the mistake, which has been now corrected in the revised manuscript. In all reactions, we have identified that there is no ice formation in water or on the water surface.

Question 5:

Diameter of glass jar used for polymer film formation is 6 cm (line 87) while the obtained film is 8 cm (line 113 and Figure 2) in diameter. Since both of these films are different, are the same conditions (concentration of monomer/oxidant *etc.*) been used for both films?

Response:

We are sorry for the confusion on the diameters of the glass wells. The wafer-sized q2D PANI film was synthesized in a glass well of 8 cm in diameter (Supplementary Fig. 2), while all of other films were prepared in the glass well of 6 cm. The confusion has been corrected in the revised manuscript. In addition, we added an additional paragraph to describe the synthesis procedures with a subtitle “**Synthesis and transfer of wafer-scale q2D PANI**” in Supplementary Methods (Page 34): “*The wafer-scale of q2D PANI was synthesized in a 150 mL crystallising dish (diameter $\varnothing = 8$ cm), and 100 mL Millipore water was used. The amounts of other reagents (e.g. aniline, APS and HCl) were doubled, while the synthesis procedures are same to above.*”

Question 6:

The conductivity enhancement of PANI seems to be predominated by HCl doping rather than the synthetic novelty of PANI in presence of surfactant layer. Supplementary Figure 20 shows very low conductivity for the 9.3 nm thick PANI films formed with 0.02 M HCl dopant in water. This is further extended to 23 S/cm using 1 M HCl in water and further to 160 S/cm under acid treatment. What concentration of HCl was used to reach conductivity up to 160 S/cm?

Response:

The HCl doping generally enhances the conductivity of PANI due to the formation of emeraldine salt via protonation. The low conductivity of q2D PANI in Supplementary Fig. 20 (now Supplementary Fig. 25) is due to the low doping level (0.02 M HCl). It is worth to note that a much higher amount of HCl (1 M) has been used in literature to dope PANI for achieving higher electric conductivity (see *Angew. Chem.* 2015, 54, 10501). Apart from this report, no conductivity data was available for the PANI thin films (especially when the film thickness was below 30 nm) prepared via spin-coating, LB method or SAM templating (Supplementary Table 2). The main reason is likely that the PANI thin films prepared by these method possess lower crystallinity and morphological homogeneity. Our

approach, on the other hand, provides direct access to a highly crystalline film with controllable thickness and wafer-scale lateral size, which can be beneficial for the practical applications of conducting PANI films. The lateral conductivity of the PANI films synthesized in this work can be tuned via intrinsic doping (0.02 M HCl with $8.7 \times 10^{-3} \text{ S cm}^{-1}$; 1 M HCl with 23 S cm^{-1}). With post-synthetic vapour-phase doping (concentrated HCl, 37%), the lateral conductivity was further enhanced to 160 S cm^{-1} .

Corresponding text in the caption of Supplementary Fig. 25 has been modified in the revision “*The sample was then doped by HCl vapour from concentrated HCl (37%) in a sealed flask for 2 h to achieve a conductivity of ca. 160 S/cm*”.

Question 7:

In the context of previous point, the highest conductivity of as-formed PANI films is found to be 23 S/cm (not 160 S/cm), which is not significantly high as compared to those existing in literature reports. For example, direct formation of pristine PANI films show excellent conductivity of 35 S/cm (*Angew. Chem.* **2015**, 54, 10501), 41 S/cm (increased to 188 S/cm by HCL doping; *Angew Chem* **2016**, 55, 12516), *etc.*

Response:

We thank the reviewer for the constructive comment and sharing these papers. Actually, the conductivity of 35 S/cm was measured on the PANI film of 30 nm thick (*Angew. Chem.* 2015, 54, 10501), and the 41 S/cm was measured on a 7.96 μm thick film prepared by the filtration of PANI nanosheets (*Angew Chem* 2016, 55, 12516). High conductivity is more challenging to achieve when the film thickness is very low (e.g. <30 nm, please also refer to our Response to Question 6 and Supplementary Table 2).

Question 8:

Is there any effect on the thickness of PANI film formation upon changing concentration of HCl in water from 0.02 to 1 M during polymerization? For example, it is stated to obtain 9.3 nm thick PANI film using 0.02 M HCl in water. Will the thickness remain same for identical reaction conditions and polymerization time while using 1 M HCl in water?

Response:

We thank the reviewer for the valuable comment. The polymerization was a bit slower (especially in early stage) when the acid concentration increased from 0.02 M to 1 M HCl. The growth kinetic of the q2D PANI at 1 M HCl is shown in Figure R9, in which a longer induction period (ca. 12 h) is observed (*Prog. Polym. Sci.* 1998, 23, 1443-1484). In addition, the polymerization speed decreases to ~4.2 nm per day. The new results have been included in the Supplementary Fig. 16. And a corresponding discussion has been included in Page 8: “*In order to increase the doping level of q2D PANI, the acid concentration increases to 1 M during polymerization, while a longer induction period (ca. 12 h) is observed and the polymerization speed decreases to 4.2 nm per day (Supplementary Fig. 15)*”

Figure R9 | Growth kinetic of the q2D PANI at 1 M HCl in water under sodium oleyl sulfate monolayer. The thickness was measured with variable angle spectroscopic ellipsometry with 5 positions on each sample.

Question 9:

The authors didn't analyze π - π stacking distance of resultant 2D PANI films. In Figure 3d, it is stated that interplanar spacing is 13.5Å (unit cell parameters, $a = 6.79$ Å, $b = 7.45$ Å, $c = 13.41$ Å, and $\alpha = 97^\circ$, $\beta = \gamma = 90^\circ$), which is a way larger than conventional value around 3.5 Å reported for many PANIs. This reviewer strongly suggests carrying out additional X-ray diffraction experiments for accurate analysis of crystalline structure. This should be done because the main point of this paper is the formation of high crystalline PANI.

Response:

We greatly appreciate the reviewer for the constructive comment. The value of interplanar spacing is 13.5Å, which is indeed much larger than normal layered materials or conjugated polymers with π - π

stacking distance of ~ 3.5 Å. Large spacing between the layers of q2D PANI arises from the intercalation of Cl^- ions. We have performed XRD measurements, but there was no clear peak which could be identified due to the ultra-thinness of PANI film. By using synchrotron grazing-incidence wide-angle X-ray scattering (GIWAXS; P03-MINAXS beamline, DESY), we obtained high quality data giving detailed unit cell parameters: $a = 6.79$ Å, $b = 7.45$ Å, $c = 13.41$ Å, and $\alpha = 97^\circ$, $\beta = \gamma = 90^\circ$ (Fig. R10 or Supplementary Figs. 12). Furthermore, from unit cell simulation (Fig. R11 or Supplementary Fig. 12), the $p2gg$ plane group symmetry is verified since the odd order (h00) and (0k0) peaks are absent in GIWAXS pattern. The unit cell obtained from GIWAXS is in excellent agreement with SAED result (Fig. 3a and 3c), and is further confirmed by DFT calculation (Supplementary Fig. 14).

Figure R10 | GIWAXS Characterization. (a) GIWAXS patterns of ca. 30 nm q2D PANI on SiO_2/Si wafer. (b) In-plane and out-of-plane projections from (a). The position of the 001 peak depends on the orientation of crystallites with respect to the substrate plane. The peak position is slightly shifted in-plane and out-of-plane, that is because of the high detector tilt angle which leads to some degree of freedom while calibration. The figures are from Supplementary Figs. 12.

Figure R11 | GIWAXS patterns of q2D PANI and unit cell simulation. The $p2gg$ plane group symmetry is verified since the odd order ($h00$) and ($0k0$) peaks are absent in GIWAXS. The figure is from Supplementary Fig. 13.

Question 10:

Ammonia sensing using PANI is not entirely novel application. PANI messes are reported to exhibit high-performance sensing of ammonia up to concentration of 2.5 ppb (*RSC Adv.* **2018**, 8, 5312).

Response:

We thank the reviewer for sharing this literature. We fully agree with reviewer that ammonium sensing is indeed not a novel application. In this work, we aimed at a proof-of-concept study to demonstrate the functional property of the achieved PANI films. We believe that the highly crystalline PANI films can have many promising applications. For example, in addition to ammonium and VOC sensing described in this work, we also found that the q2D PANI film acting as an active layer showed excellent specific capacitance and energy density in thin-film microsupercapacitor (MSC). This result will be reported in a separate work.

Therefore, in order to avoid the overwhelming claim, the corresponding text regarding sensing performance has been modified in the revised manuscript as follows:

The “*outperforming classical linear PANI counterparts*” in abstract and “*rivaling the state-of-art PANI counterparts*” in Page 3 Line 10 have been removed.

The “*lower than other reported PANI sensors (Fig. 5b and Supplementary Table 3)*” in Page 10 the last Line has been modified to “*lower than the most reported PANI sensors (Fig. 5b and Supplementary Table 3)*”

Question 11:

While the authors tried to apply their materials in VOC sensing applications, as can be seen in Figure 5, the resolution and stability do not meet current technology. In fact, the resistance keeps increasing with sensing, indicative of non-complete desorption of ammonia from PANI thin films.

Response:

Regarding the VOC sensor (i.e. heptanal, Fig. 5c and 5d), we have demonstrated that the sensitivity is superior to the state-of-art PANI based device (~1.7 % at 25 ppm; *Adv. Healthc. Mater.* 2018, 7, 1800232). Such sensitivity is sufficient to identify the compounds released from patients of lung cancer (~205.5 ppm) and healthy controls (~22.8 ppm) (*Chem. Rev.*, 2012, 112, 5949; *Nat. Nanotechnol.* 2009, 4, 669). Regarding ammonia sensing, the q2D PANI based chemiresistor could detect NH₃ gas as low as 30 ppb ($\Delta R/R_0 \approx 1.8\%$), which is superior to most of the reported PANI-based sensors (Supplementary Table 3). Using the same testing setup, our sensitivity and stability are even better than nitrogen- and boron-doped carbon nanotubes (100 ppb; *ACS Sens.* 2018, 3, 79). Such sensitivity is also sufficient for diagnosis of certain diseases such as liver cirrhosis, kidney failure, and diseases caused by *Helicobacter pylori* (*Dig. Dis. Sci.* 2002, 47, 2523). The application of q2D PANI in sensing is a kind of proof-of-concept demonstration in the current work. Considering the criticism from the reviewer, and corresponding text regarding sensing performance have been modified in the revised manuscript (shown in the Response to the Question 10).

Reviewer #3 (Remarks to the Author):

Comment:

General Comments: This paper reports a simple method using air-water interface and surfactant monolayer as templates to synthesize crystalline quasi-two-dimensional (q2D) PANI and It can be tunable thickness (2.6-30 nm). In addition, the authors demonstrate that crystal structure quasi-two-dimensional (q2D) PANI by grazing-incidence wide-angle X-ray scattering (GIWAXS) and Aberration-corrected high-resolution transmission electron microscopy (AC-HRTEM). Due to high crystallinity, q2D PANI exhibits ultra-high lateral conductivity up to 160 S cm⁻¹ doped by HCl and superior chemiresistive sensing toward ammonia as low as 30 ppb, and volatile organic compounds (e.g. heptanal) at 10 ppm. However, The authors needs to explains the relationship between tunable thickness and crystallinity. Besides, the manuscript is strewn with poor usage of the English language. Therefore, I recommend publication of the manuscript with a MAJOR REVISION, in light of the following specific comments:

Response:

We greatly appreciate the reviewer's valuable comments on the work. Regarding the relationship between film thickness and crystallinity, we have performed SAED measurements on the q2D PANI films with varied thicknesses collected at different reaction times (from 12 h to 7 days; Figure R4). The crystal structures of the films are always identical at various thicknesses. However, the crystallinity increased continuously and obviously during the reaction/polymerization, since the SAED patterns became clearer and brighter. Other substantial concerns that reviewer raised have been carefully addressed as follow.

Question 1:

The authors stated that free oligomers and polymers (in solution) that cannot interact with the surfactant monolayer will precipitate. Figure 1 (a) also shows this content. Then it means that there is a limit to film formation? In other words, The authors should explain exactly principle of the thickness control of the film.

Response:

We thank the reviewer for the constructive comments. The limit in the thickness of the q2D PANI film of the current conditions is ca. 30 nm as shown in Fig. 4a. It has been reported that the polymerization of aniline can be strongly affected by the concentrations of monomer, oxidant and acid, as well as the reaction temperature. (*Prog. Polym. Sci.* 1998, 23, 1443). Thus, the thickness of the PANI film can also be affected by these parameters. In the current work, we mainly focused on the synthesis method and corresponding structural analysis of the resulting q2D PANI films. Therefore, the synthetic conditions (such as the concentrations of monomer and oxidant, and reaction temperature) were fixed. We have demonstrated that the thickness of the PANI film increased as a function of polymerization time (Fig. 4a). Longer polymerization time provided thicker q2D PANI. To be specifically, the thickness of q2D PANI increased with a constant rate of $\delta d = 5$ nm per day in the first five days, and then levelled around 30 nm after seven days when all monomers were consumed. Regarding the reviewer's concern, we have modified the text in Page 8 Line 18 and the new version is "*The q2D PANI formation is confined at the surfactant-water interface, in which the monomers in the water subphase continuously transport to the interface for oxidative polymerization. Therefore, extending the reaction time leads to a higher monomer conversion, corresponding to an increase in film thickness*".

Question 2:

The authors stated that low concentrations of monomer are beneficial. Proper reasons should be presented. Do you also control the concentration of monomer for thickness control of the film? If so, why is there a precipitated polymers?

Response:

We thank the reviewer for the thoughtful comment. The reason for using low concentrations of monomer is to decrease the rates of nucleation, polymerization and crystal growth, and to finally achieve the synthesis of ultra-thin PANI film with excellent crystallinity. In initial experiments, we have tried different monomer concentrations, and found that the resultant PANI film was less uniform when the monomer concentration was too low (< 0.06 mM in 50 mL water). But the precipitation occurred in all cases, since both monomers and oxidants (ammonium persulfate) coexisted in the

water bath, although the amount of precipitated polymers decreased significantly when the concentration of monomer decreased. Regarding the comment, we have modified the text in Page 3 Line 22 and the new version is “*The polymerization was slowed down by using low concentration of monomer and oxidant (i.e. APS), which could be favorable for the formation of ultra-thin PANI films with high crystallinity*”.

Question 3:

After addition of aniline into water sub phase, The authors add hydrochloric acid and ammonium persulfate after 24h. What is the clear reason to put in after 24 hours? Proper reasons should be presented.

Response:

In the previous experiments, we found that the reagents (monomers and oxidant) needed at least 15 h to diffuse sufficiently in the whole glass well at 1 °C. Therefore, the reason for the addition of oxidant after 24 h was to make sure that aniline monomers were evenly distributed in the glass well as well as adsorbed on the interface of the surfactant monolayer. To elucidate our motif more clearly, a sentence “*standing for 24 h for the diffusion of monomers to the water subphase and interface*” has been added in the caption of Fig. 1a as well as Page 34 Line 12 of Supplementary Methods.

Question 4:

After 48h, Atomic force microscopy (AFM) image show homogeneity film thickness. Then is film thickness still uniform and homogeneity after 7 days? In addition, The authors have to show whether optical microscopy images also appear uniform as a function of time, In other words, The authors should explain that film has uniform and continuous in various film thickness through Atomic force microscopy (AFM) image or optical microscopy image.

Response:

We thank the reviewer for the constructive comment. Following the reviewer’s suggestion, we have measured the morphologies of q2D PANI films of various thickness/polymerization time by AFM and optical microscopy. As shown in Figure R12, regardless of thickness, the resulting q2D PANI films were uniform and homogeneous at microscopic scale. We observed only an insignificant increase in

surface roughness when the film thickness reached 19.6 (RMS = ~ 0.8 nm) and 30.8 nm (RMS = ~ 0.9 nm), respectively. Nevertheless, the morphologies are in clear contrast to that of prepared in the absence of surfactant monolayer (Fig. R2a), where the resultant PANI film is rather rough and discontinuous.

Figure R12 | Morphology characterization of q2D PANI films of various thicknesses. (a-d) Optical microscopy images of q2D PANI films on SiO₂ substrates. (e-d) AFM topographic images and high profiles of the q2D PANI films corresponding to (a-d), respectively. Scale bars: (a-d) 20 μm ; (e) 3 μm ; (f) 10 μm ; (g) 20 μm ; (h) 15 μm .

Question 5:

In Figure 1, The authors have to make a clear notation in the picture. Make it clearly what is the oligomers, polymer, APS, HCl and so on.

Response:

We thank the reviewer for the constructive comment. Following the suggestion, we have added the notations in Fig. 1a in the revised manuscript.

Question 6:

In Figure 3, AC-HRTEM image is appeared clearly, but check again Scale bars. (b) and (d) represent 5nm and 10nm respectively. Is it correct? Then The authors should show low magnification of TEM image formed 2D single crystal.

Response:

We thank the reviewer for the valuable comment. We have double checked the scale bars in Fig. 3b and 3d, and they were correct. The lattice distances (3b: in-plane; 3d: out-of-plane) showing the HRTEM images match well with the unit cell parameters obtained from SAED and GIWAXS. The low magnification TEM images of q2D PANI (with the thickness of 2.6 nm and 9.3 nm) were shown in Supplementary Fig. 10, 11 and 21 in our previous version. Following the advice from the reviewer, in Fig. R13, we provide an additional low magnification of TEM image of 30-nm q2D PANI film. Unfortunately, the PANI single crystal could not be identified by TEM imaging.

Figure R13 | Low magnification TEM image of q2D PANI of 30 nm thick.

Question 7:

The manuscript requires a complete overhaul of the language (grammar, missing articles, inappropriate phrasing, etc.). The authors are strongly encouraged to take the support of native language editors for the purpose.

Response:

We thank the reviewer for the constructive comment. The manuscript has been further polished by different co-authors and native speakers. All the changes in the manuscript are highlighted with a yellow background.

REVIEWERS' COMMENTS:

Reviewer #1 (Remarks to the Author):

In the revised manuscript, the authors did an excellent job of addressing the comments. All my previous concerns are thoroughly addressed with adequate support of additional experiments and/or discussions. The quality of the manuscript is largely improved. This work brings to the community a new strategy of creating quasi 2D organic materials with impressive physical properties, which deserves a rapid publication with high visibility. I therefore recommend publication of this work in its current form.

Reviewer #2 (Remarks to the Author):

Reviewer #2.

The authors have modified the manuscript with experimental details and relevant explanations for better understanding. The current version of manuscript also needs very sincere effort to eliminate the typos throughout the text (a few of notable errors are "concertation" line 193, "Figure caption in S17", etc). Apart from those the other points based on the original comments are listed bellow.

Original comment 1: The query was to clarify whether q2D PANI film formation is confined at (1) surfactant–water interface or (2) air–water interface. However, the authors have stated both of them to occur in the revised version. First, a surfactant monolayer is formed at air–water interface followed by migration of aniline to underneath the surfactant layer and get oxidized to make PANI.

Original comment 2: Solvents like ethanol and chloroform are usually known to deteriorate the conductivity (and capacitance) but improve the cycling stability of polyaniline. Possible change of solvent to eliminate the surfactant may result in high conductivity of as-formed q2D PANI films.

Original comment 4: This reviewer suggests to include the instrumentation and detailed procedure used to maintain the temperature exactly at 1 °C throughout the experiment for clarity.

Original comment 6: This reviewer suggests to change the statement regarding record-high conductivity for PANI thin films up to $160 \text{ S}\cdot\text{cm}^{-1}$. It is clear that high-conductivity is achieved after external treatment rather than direct synthesis of 2D PANI (highest conductivity $23 \text{ S}\cdot\text{cm}^{-1}$ with 1M HCl).

Reviewer #3 (Remarks to the Author):

The authors have had enough responses to reviewers' comments. In particular, the authors showed the crystal structure of quasi-two-dimensional (q2D) PANI by grazing-incidence wide-angle X-ray scattering (GIWAXS) and Aberration-corrected high-resolution transmission electron microscopy (AC-HRTEM). DOI: 10.1038/ncomms12803, DOI: 10.1039/c7nr01060e should be cited for related examples of crystal structure of conducting polymer.

Reviewer #1 (Remarks to the Author)**Comments:**

In the revised manuscript, the authors did an excellent job of addressing the comments. All my previous concerns are thoroughly addressed with adequate support of additional experiments and/or discussions. The quality of the manuscript is largely improved. This work brings to the community a new strategy of creating quasi 2D organic materials with impressive physical properties, which deserves a rapid publication with high visibility. I therefore recommend publication of this work in its current form.

Response:

We thank the reviewer for the positive comment on our revised manuscript.

Reviewer #2 (Remarks to the Author)**Comments:**

The authors have modified the manuscript with experimental details and relevant explanations for better understanding. The current version of manuscript also needs very sincere effort to eliminate the typos throughout the text (a few of notable errors are “concertation” line 193, “Figure caption in S17”, etc). Apart from those the other points based on the original comments are listed bellow.

Response:

We thank the reviewer for the positive comments on the revised manuscript. We are sorry for these errors, which have been corrected in the revision. Other text and captions have been checked and modified accordingly. The additional comments from the reviewer have been addressed as following.

Comment 1:

Original comment 1: The query was to clarify whether q2D PANI film formation is confined at (1) surfactant–water interface or (2) air–water interface. However, the authors have stated both of them to

occur in the revised version. First, a surfactant monolayer is formed at air–water interface followed by migration of aniline to underneath the surfactant layer and get oxidized to make PANI.

Response:

We are sorry that we have insufficiently understood the reviewer’s Comment 1. We agree with reviewer’s judgement that the surfactant monolayer is firstly formed at air-water interface and the aniline monomer is absorbed and polymerized underneath the surfactant monolayer, which has been depicted in Fig. 1a of the manuscript.

Original comment 2: Solvents like ethanol and chloroform are usually known to deteriorate the conductivity (and capacitance) but improve the cycling stability of polyaniline. Possible change of solvent to eliminate the surfactant may result in high conductivity of as-formed q2D PANI films.

Response:

We appreciate the reviewer for the valuable suggestion on sample cleaning. On the motif of both fundamental study and device application, we are eager to further improve the conductivity of the q2D PANI films. In our experiment, the conductivities were decreased by 10-20% after cleaning. All conductivity values used in the manuscript are obtained on cleaned sample. The solvent effect that reviewer mentioned is interesting to further increase the conductivity, thus we will closely investigate it in following work.

Original comment 4: This reviewer suggests to include the instrumentation and detailed procedure used to maintain the temperature exactly at 1 °C throughout the experiment for clarity.

Response:

Following the reviewer’s suggestion, we have included the detailed procedure and instrumentation for controlling the temperature in our synthesis. A modified sentence has been included in the Method section (Page 13 Line 7): *“The glass well was then covered by a glass slide and placed in a refrigerator (Liebherr FKUv 1660 Premium, Germany) at 1 °C for the oxidative polymerization.”*

Original comment 6: This reviewer suggests to change the statement regarding record-high conductivity for PANI thin films up to $160 \text{ S}\cdot\text{cm}^{-1}$. It is clear that high-conductivity is achieved after external treatment rather than direct synthesis of 2D PANI (highest conductivity $23 \text{ S}\cdot\text{cm}^{-1}$ with 1M HCl).

Response:

We thank the reviewer for the constructive comment. The words of “record-high” about conductivity has been removed in the revised manuscript.

Corrections made:

The word “*record-high*” was removed in the third line of Discussion.

The word “*high*” was removed in the last third line of Abstract.

Reviewer #3 (Remarks to the Author):

Comments:

The authors have had enough responses to reviewers' comments. In particular, the authors showed the crystal structure of quasi-two-dimensional (q2D) PANI by grazing-incidence wide-angle X-ray scattering (GIWAXS) and Aberration-corrected high-resolution transmission electron microscopy (AC-HRTEM). DOI: 10.1038/ncomms12803, DOI: 10.1039/c7nr01060e should be cited for related examples of crystal structure of conducting polymer.

Response:

We appreciate the reviewer for the positive comment on our revised manuscript. Following the reviewer's suggestion, we have added the two new references (Ref. 10 and 11) in the revision.